# MUG Mel3 Cell Lines Reflect Heterogeneity in Melanoma and Represent a Robust Model for Melanoma in Pregnancy

**DOI:** 10.3390/ijms222111318

**Published:** 2021-10-20

**Authors:** Silke Schrom, Thomas Hebesberger, Stefanie Angela Wallner, Ines Anders, Erika Richtig, Waltraud Brandl, Birgit Hirschmugl, Mariangela Garofalo, Claudia Bernecker, Peter Schlenke, Karl Kashofer, Christian Wadsack, Ariane Aigelsreiter, Ellen Heitzer, Sabrina Riedl, Dagmar Zweytick, Nadine Kretschmer, Georg Richtig, Beate Rinner

**Affiliations:** 1Division of Biomedical Research, Medical University of Graz, 8036 Graz, Austria; silke.schrom@medunigraz.at (S.S.); thomas.hebesberger@medunigraz.at (T.H.); stefaniewallner@a1.net (S.A.W.); ines.anders@medunigraz.at (I.A.); 2Department of Dermatology, Medical University of Graz, 8036 Graz, Austria; erika.richtig@medunigraz.at; 3Department of Obstetrics and Gynecology, Medical University of Graz, 8036 Graz, Austria; waltraud.brandl@medunigraz.at (W.B.); birgit.hirschmugl@medunigraz.at (B.H.); christian.wadsack@medunigraz.at (C.W.); 4BioTechMed-Graz, 8010 Graz, Austria; sabrina.riedl@uni-graz.at (S.R.); dagmar.zweytick@uni-graz.at (D.Z.); 5Department of Pharmaceutical and Pharmacological Sciences, University of Padova, 35122 Padova, Italy; mariangela.garofalo@unipd.it; 6Department of Blood Group Serology and Transfusion Medicine, Medical University of Graz, 8036 Graz, Austria; c.bernecker@medunigraz.at (C.B.); peter.schlenke@medunigraz.at (P.S.); 7Diagnostic and Research Institute of Pathology, Medical University of Graz, 8036 Graz, Austria; karl.kashofer@medunigraz.at (K.K.); ariane.aigelsreiter@medunigraz.at (A.A.); 8Institute of Human Genetics, Diagnostic and Research Center for Molecular BioMedicine, Medical University of Graz, 8036 Graz, Austria; ellen.heitzer@medunigraz.at; 9Institute of Molecular Biosciences, Biophysics Division, University of Graz, 8010 Graz, Austria; 10BioHealth, 8010 Graz, Austria; 11Institute of Pharmaceutical Sciences, Department of Pharmacognosy, University of Graz, 8010 Graz, Austria; nadine.kretschmer@uni-graz.at; 12Division of Oncology, Medical University of Graz, 8036 Graz, Austria; georg.richtig@medunigraz.at

**Keywords:** melanoma, tumor heterogeneity, pregnancy, anti-tumor peptides, in vitro model

## Abstract

Melanomas are aggressive tumors with a high metastatic potential and an increasing incidence rate. They are known for their heterogeneity and propensity to easily develop therapy-resistance. Nowadays they are one of the most common cancers diagnosed during pregnancy. Due to the difficulty in balancing maternal needs and foetal safety, melanoma is challenging to treat. The aim of this study was to provide a potential model system for the study of melanoma in pregnancy and to illustrate melanoma heterogeneity. For this purpose, a pigmented and a non-pigmented section of a lymph node metastasis from a pregnant patient were cultured under different conditions and characterized in detail. All four culture conditions exhibited different phenotypic, genotypic as well as tumorigenic properties, and resulted in four newly established melanoma cell lines. To address treatment issues, especially in pregnant patients, the effect of synthetic human lactoferricin-derived peptides was tested successfully. These new *BRAF*-mutated MUG Mel3 cell lines represent a valuable model in melanoma heterogeneity and melanoma pregnancy research. Furthermore, treatment with anti-tumor peptides offers an alternative to conventionally used therapeutic options—especially during pregnancy.

## 1. Introduction

Melanoma is one of the most aggressive and heterogeneous types of cancers, with a strong tendency to develop resistance against different therapy approaches [1,2]. Among cutaneous melanomas, approximately 35% are diagnosed during gestation or the postpartum period, forming a specific subset of patients with pregnancy-associated melanoma (PAM) [3]. This can be due to changing life situations; more women are having children at an older age and the incidence of pregnancy-related cancers increases generally with age [4].

Another challenge in the cure of melanoma is that melanoma presents with one of the highest mutation rates of all cancers [5]. In 80% of all cases, mutations occur within the RAS-RAF-MEK-ERK-MAP kinase pathway [6]. The *BRAF* gene is mutated with a hotspot mutation at codon 600 (mostly V600E) in 40% to 50% of all melanoma cases, followed by NRAS mutations (G12, G13, Q61) in 12% to 20% [7]. However, to date, FDA-approved targeted therapies in advanced cutaneous melanoma have been available only for *BRAF* mutations [8]. Several years ago, *BRAF* inhibitors such as vemurafenib, dabrafenib, and encorafenib revolutionized the treatment of melanoma, nowadays given in combination with MEK inhibitors like cobimetinib, trametinib and binimetinib. In addition, immunotherapeutic agents like the anti-CTLA-4 antibody ipilimumab and the anti-PD-1 antibodies pembrolizumab and nivolumab have improved disease outcomes in melanoma patients [9,10]. Melanoma responds well to targeted therapies and immune therapies; however, resistance occurs in a significant percentage of patients [1,11]. 

Consequent to tumor heterogeneity in advanced melanoma, resistances against therapies for *BRAF* and MEK inhibitors develop. Intratumoral heterogeneity therefore has important implications for treatment selection [12]. The relationship between genomic and immunological heterogeneity, as well as the link to tumor growth and therapy response, has not been extensively studied [13]. 

Furthermore, due to pigmentation, melanomas exhibit a diverse morphology. To date, the role of melanin in melanoma, particularly its effects on the ability to metastasize, is almost unknown. Sarna et al. showed in mouse experiments that increased melanin expression inhibits the invasive capabilities of melanoma cells in vitro based on cell elasticity and summarized that cells with melanin expression were able to spread less than non-pigmented cells [14].

To better understand the biology of melanoma and thus develop new therapies specifically targeted for pregnant patients, it is necessary to establish patient-derived cell line models that reflect the heterogeneity of melanoma. Most existing melanoma cell lines are cultured in fetal bovine serum (FBS) and result in one cell line. This only reflects a small clonal proportion of the heterogenic tumor cell pool. Human platelet lysate (hPL), an alternative to FBS, is rich in several nutrients that promote cell growth and proliferation. However, it is currently not widely used in cell culture [15,16]. 

In particular, the treatment of cancers during and shortly after pregnancy is a challenge. Two peptides: R-DIM-P-LF11-322 and R-DIM-P-LF11-334—derived from the human host defense peptide lactoferricin (hLFcin)—were used for innovative melanoma treatment. Both peptides were previously shown to exert antitumor activity on different melanoma cells in vitro and in vivo [17,18,19,20,21]. Riedl et al. were able to prove that the negatively charged phospholipid phosphatidylserine (PS) is exposed on the outer leaflet of the plasma membrane of melanoma cells [22,23,24]. This general marker of cancer cells can serve as a target for host defense peptides like R-DIM-P-LF11-322 and R-DIM-P-LF11-334, which have emerged as a potential new and alternative anti-cancer therapeutic [18,21]. 

In this study, we present a patient with pregnancy-associated lymph node metastasis (PALNM) from a *BRAF*-mutated melanoma. Thereby, the aim was to establish multiple well-characterized cell lines from one patient, preserving intratumoral heterogeneity. A pigmented and non-pigmented tumor fraction were considered, and different cultivation conditions were used. Lastly, the effects of innovative peptides for melanoma treatment, such as R-DIM-P-LF11-322 and R-DIM-P-LF11-334—which might be of particular interest for pregnant patients—were assessed.

## 2. Results

### 2.1. Cell Growth and Classical Melanoma Markers Confirm the Heterogenic Character of MUG Mel3 Cell Lines

The patient received no therapy nearly one year before re-developing a new lymph node metastasis during gestation. Cells from this lymph node metastasis were used to establish MUG Mel3 cell lines. After mechanical tumor dissociation, the cells were cultured under two different growth conditions: hPL and FBS, illustrated in Figure 1.

Cells with hPL started adhering to the cell culture flask’s surface on the first day of cultivation, FBS cultures on the second day. Morphologically, cultures with hPL supplementation presented a semi-adherent growth behaviour at the start of primary cultures, marked by a more round-shaped phenotype and a high proportion of cells in suspension compared to well-adherent spindle-shaped cells in FBS cultures. No proliferation assays and statistical evaluation could be performed at the beginning of cultures due to insufficient amounts of cells, which is common in primary cell culture. Cells isolated from the pigmented part displayed a brownish staining immediately after the start of culture—especially in the hPL culture. However, this staining vanished after more than three passages. During the first two days, the focus of cultivation was to remove dead cells by centrifugation and media changes. On the third day, the first cell growth could be observed, and on day six and eight, representative pictures of the different conditions of primary cultures were taken (Figure 2). 

Proliferation was recorded by passaging, starting at day 10 for MUG Mel3 PF, on day 13 for MUG Mel3 Ph, day 23 for MUG Mel3 NPF, and day 38 for MUG Mel3 NPh. Nomenclature of the pigmented part resulted only from first impressions of the tissue sample, since no pigmentation was visible after passaging the cells. Optimal hPL concentration was determined after several months in culture, as soon as enough cells were available for analysis by MTS cell proliferation assay. The highest OD values, which indicated the highest number of viable cells, were achieved between concentrations of 1.25% and 5% hPL (Appendix A). 

MUG Mel3 Ph and MUG Mel3 NPh were then cultured in a 2.5% hPL medium. As soon as continuous growth was observed, ICC staining was performed. The melanoma marker MCSP was highly expressed in all four cell lines and did not markedly differ between the four treatment conditions. 

However, differences could be observed for melanoma markers between the pigmented and the unpigmented sections. MUG Mel3 PF stained weak for HMB45, Melan-A, and tyrosinase. MUG Mel3 Ph stained weakly for tyrosinase as well, but contrary to MUG Mel3 PF, strongly for HMB45 and medium for Melan-A. Both MUG Mel3 NPF and MUG Mel3 NPh highly expressed each of the stained melanoma markers HMB45, Melan-A, Tyrosinase, and MCSP (Figure 3, Table 1). 

Melan-A and SOX10, a marker used to identify metastatic melanoma [25], were assessed by qPCR, demonstrating significantly higher values for hPL cultured cells compared to FBS supplemented cells. Normalized to MUG Mel3 NPF (1 ± 0.11-fold expression), SOX10 mRNA was 0.93 ± 0.02-fold expressed in MUG Mel3 PF, 1.58 ± 0.76 in MUG Mel3 NPh, and 1.5 ± 0.24 in MUG Mel 3 Ph. Comparing cells of the same origin based on pigmentation but different cultivation methods, the analysis revealed that cells cultured from the pigmented part had a 1.6-fold higher SOX10 mRNA expression when established with hPL supplementation (*p* = 0.015) and a 1.58-fold higher expression for the non-pigmented part (*p* = 0.258). Melan-A mRNA expression was even more varied, with a 0.24 ± 0.3-fold expression of MUG Mel3 PF, 2.64 ± 0.3 of MUG Mel3 NPh, and 1.71 ± 0.26 of MUG Mel3 Ph normalized to MUG Mel3 NPF with 1 ± 0.34.

The mRNA expression was greater for the pigmented and non-pigmented parts in hPL cultures, with a 7.26-fold higher expression for the pigmented part (*p* = 0.003) and a 2.64-fold higher expression for the non-pigmented part (*p* = 0.002). Moreover, we found significant differences in Melan-A mRNA expression levels when comparing cells derived from the pigmented and the non-pigmented part of the metastasis. The non-pigmented part had higher Melan-A expression levels independently of hPL or FBS supplementation, and was 4.25-fold higher in FBS cultures (*p* = 0.04) and 1.55-fold higher in hPL cultures (*p* = 0.01). However, SOX10 mRNA expression did not differ (for FBS cultivation *p* = 0.364 and for hPL cultured cells *p* = 0.859; Figure 4A,B, Table 1). Furthermore, CD271—a stemness marker associated with cell migration properties in melanoma—was measured in biological triplicates by flow cytometry. Different cell culture supplementation with either hPL or FBS did not affect CD271 expression. A higher percentage of CD271+ cells could be observed for cells derived from the pigmented part (Figure 4C). 

STR authentication was carried out for all four cell lines and confirmed origins of the pigmented and the non-pigmented part, with differences for both parts at locus D16S539. Losses could be observed for higher passages at locus D16S539 for MUG Mel3 NPh and at locus D7S820 for MUG Mel3 Ph (Appendix A).

### 2.2. Measurement of Growth Factors and Chemokines with Luminex Technology

Growth factors were measured at the very beginning of primary cell culture, before testing optimal amounts of hPL on cell proliferation. Therefore, supernatant from cells cultivated in RPMI supplemented with either 10% FBS or 10% hPL were compared for human chemokine expression (Appendix A). Melanoma cells of the non-pigmented part, cultured with FBS or hPL, both secreted growth-regulated oncogene (GRO) alpha (FBS: 501.32 ± 15.66 pg/mL and hPL: 1172.46 ± 37.56 pg/mL) and IL-8 (FBS: higher than standard 1 at 1050 pg/mL and hPL: 885.83 ± 27.73 pg/mL) into the supernatant. Whereas RANTES was only excreted by MUG Mel3 supplemented with hPL (815.51 ± 100.45 pg/mL; Figure 4D). 

### 2.3. Copy Number Profiling

To compare the copy number status of the cell lines and the primary tissues, a sWGS was performed. To this end, genome-wide copy number profiles were established from the pigmented and non-pigmented tissues as well as for two different passages of the FBS and hPL cultured cell lines (Figure 5). 

While some CNAs, such as losses at chr6q, chr9p, chr9q, chr10, chr11p, or gains at chr3q, chr7q, and chr8q were shared by both tumor tissue and cell lines—indicating a common origin of all samples—other CNAs (gains on chr2, 3,17,18) were observed only in a subset of samples. Interestingly, some changes (in particular on chr3) seemed to occur only after culturing and were not detected in either of the tissue samples. Cell lines derived from the non-pigmented tissue shared more CNAs with each other as compared to the pigmented tissue, with only a few differences between the FBS and hPL cultured cells. Moreover, CNAs were more stable in the non-pigmented tissue over passages regardless of supplementation with either FBS or hPL (Figure 5). 

### 2.4. BRAF Mutation Analysis

A next-generation sequencing (NGS) analysis of pathologic lymph nodes in December 2018 revealed a *BRAF* V600E mutation (*BRAF*:NM_004333:exon15:c.T1799A:p.V600E) with variant allele frequencies (VAF) of 63.66 and 60.77% for the two replicates. No other mutation was identified in the analysed regions. *BRAF* V600E was assessed in pigmented and non-pigmented parts of the tissue as well as in the derived cultures using ddPCR. VAFs did not significantly differ between any of the samples with VAFs of 77.01% for the pigmented tissue, 73.98% for the non-pigmented tissue, 79.90% for MUG Mel3 PF, 85.24% for MUG Mel3 NPF, 79.85% for MUG Mel3 Ph, and 85.97% MUG Mel3 NPh.

### 2.5. Assessment of Tumorigenicity

To directly compare the tumorigenic potential of all four cell lines, we subcutaneously injected each of MUG Mel3 NPF, MUG Mel3 NPh, MUG Mel3 PF, and MUG Mel3 Ph into mice of two different strains. Thereby, we used CR ATH HO lacking T cells and NGX mice deficient in T cells, B cells, and natural killer cells. Twenty-five days post injection, all MUG Mel3 cell lines formed tumors, whereas cells derived from the non-pigmented part presented a higher tumor growth rate compared to cells derived from the pigmented part (Table 2 and Figure 6).

MUG Mel3 Ph recovered pigmentation in vivo (Figure 6C and Appendix A). IHC staining for the melanoma markers HMB45, Melan-A and tyrosinase were conducted and are depicted in Table 2. HMB45 and Melan-A were expressed in all xenograft tumors to a different extent. Tyrosinase expression could only be observed in hPl cultivated cells (Table 2, Appendix A). 

### 2.6. PS Exposure and Peptide Treatment

Since new innovative therapies with low side effects are sought, and as the anti-tumor peptides used in this study have recently been tested promisingly on melanoma cell lines in vitro, they might represent a suitable approach for the treatment of pregnant women. PS exposure, serving as a target for anti-cancer peptides like R-DIM-P-LF11-322 and R-DIM-P-LF11-334, was quantified for MUG Mel3 cells using the Annexin V (AV) apoptosis kit. Cells cultivated with FBS presented with significantly (MUG Mel3 PF/MUG Mel3 NPF: *p*-value = 0.0250; MUG Mel3 PF/MUG Mel3 Ph *p*-value = 0.0002; MUG Mel3 NPh/ MUG Mel3 Ph: *p*-value = 0.0887 and finally MUG Mel3 NPF/MUG Mel3 NPh *p*-value = 0.0013) more PS on the outside of cells in both cultured parts, pigmented and non-pigmented (Appendix A). The toxicity of anti-cancer peptides was measured by PI-uptake. Both tested peptides, R-DIM-P-LF11-322 and R-DIM-P-LF11-334, were also highly active against MUG Mel3 cells (Table 3).

Comparing both peptides, R-DIM-P-LF11-322 elicited cytotoxic effects at lower concentrations than R-DIM-P-LF11-334, ranging from 4.3 ± 0.3 µM for MUG Mel3 Ph up to 22.1 ± 1.2 µM for MUG Mel3 NPF. When focusing on the origin of cells, IC_50_-values for R-DIM-P-LF11-322 and R-DIM-P-LF11-334 on cells cultured from the non-pigmented part of the lymph node metastasis were significantly higher compared to cells from the pigmented part, as analysed by the student’s t test (*p* < 0.001, except the effect from R-DIM-P-LF1134 in MUG Mel3 PF/ MUG Mel3 NPF (*p* < 0.01)). In detail, the effect of R-DIM-P-LF11-322 compared for all cell lines: MUG Mel3 PF/ MUG Mel3 Ph *p*-value = 0.0001, MUG Mel3 PF/ MUG Mel3 NPF *p*-value = 0.0005; MUG Mel3 NPF/MUG Mel3 NPh *p*-value = 0.0021 and MUG Mel3 NPh/MUG Mel3 Ph *p*-value = 0.0001 and, in detail, the effect of R-DIM-P-LF11-334 compared for all cell lines: MUG Mel3 PF/ MUG Mel3 Ph *p*-value= 0.0001; MUG Mel3 PF/ MUG Mel3 NPF *p*-value = 0.0074, MUG Mel3 NPF/ MUG Mel3 NPh *p*-value = 0.232, MUG Mel3 NPH/MUG Mel3 Ph *p*-value = 0.0001 (Figure 7A–D). 

## 3. Discussion

Malignant melanoma is still one of the most aggressive and therapy-resistant tumors, caused by its highly intra- and intertumoral heterogenic character [1,26]. Furthermore, melanoma is one of the most prevalently diagnosed neoplasms in gestating women and is a major contributor to maternal morbidity and mortality. PAM is associated with 17% higher mortality compared with melanoma in non-pregnant female patients [27]. The incidence of melanoma development during pregnancy is increasing, largely related to an advanced maternal age [28,29]. Hyperpigmentation and increased melanocytic activity associated with melasma, as well as linea nigra, genital and areolar darkening, has been observed in pregnant women [30]. These conditions appear to correlate with elevated levels of oestrogen, progesterone, beta-endorphin, and beta- and alpha-melanocyte stimulating hormone [31]. For pregnancy-associated melanoma research, a melanoma cell line isolated directly from a pregnant patient, if possible without prior therapy, would be valuable. To our knowledge, no such cell lines are available for research currently. Research on biological mechanisms that foster PALNM and PAM could shed light on melanoma pathogenesis, leading to new therapeutic opportunities that may not only be related to pregnant patients. Particular emphasis was also placed on maintaining heterogeneity in our established cell lines, a major contributor associated with therapy resistances. 

Previous studies have identified melanoma as the most heterogeneous type of tumor, consisting of multiple distinct subclonal populations of tumor cells [2,32,33]. Our data reflect melanoma tumor heterogeneity by expression of different genotypic and phenotypic characteristics for all four established MUG Mel3 cell lines, confirmed by qPCR data and chemokine expression. From the starting point of primary cultures, a distinct growth behaviour could be observed. Cell lines cultivated under hPL conditions presented a slower proliferation rate with a semi-adherent character. Staining for Melan-A, HMB45 and tyrosinase illustrated a diverse expression pattern in cell cultures and xenografts of all four MUG Mel3 cell lines. The only ICC marker uniformly expressed in culture was MCSP, a marker abundantly expressed in most human melanoma lesions, which has been associated with melanoma cell invasion [34,35].

Furthermore, mRNA levels for SOX10, a marker for melanoma sentinel lymph node metastasis [25], displayed higher levels in hPL supplemented culture. The same trend could be observed for mRNA levels of Melan-A, which were significantly higher expressed in hPL cultured cells as well. Furthermore, GRO alpha, initially identified by its growth stimulatory activity on melanoma cells as well as RANTES, which was described as being released in high amounts by melanoma cells in the presence of FBS, were secreted in larger quantities in hPL cultivated cells [36]. In murine xenograft models Payne and Cornelius were able to demonstrate that expression of RANTES in melanoma cells formed concentration-dependent aggressive tumors in nude mice and promoted tumor progression [37]. These findings correlate with our data, as MUG Mel3 NPh, the cell line excreting the highest RANTES levels, also illustrated the highest tumorigenic potential in both tested mouse models. Except for MUG Mel3 Ph, which caused engraftment in only 80% of mice, all other three cell lines induced 100% tumor engraftment. MUG Mel3 Ph was the only cell line that also caused re-pigmentation in the majority of formed tumors. 

Contradictorily, CD271, a marker to detect stem cell-like melanoma progenitor cells, was significantly more highly expressed in cell lines derived from the pigmented part. CD271+ melanoma cells has been described as contributing to proliferation, tumorigenicity and plasticity [38,39,40], which does not coincide with our observations. This discrepancy could be explained by the previous work of Quintana et al., focusing on the usability of human melanoma cell lines. They compared genes expressed in commonly used cell lines to tumor tissues of origin and stated that the tumorigenic capacity in melanoma is not restricted to a small population. More likely, tumorigenic competence might reflect a reversible state in melanoma. They also concluded that so far, no markers are available that robustly distinguish tumorigenic from non-tumorigenic melanoma cells [41].

Regarding heterogeneity in melanoma, therapy-refractory subpopulations can cause tumor recurrence even in responding patients [42,43]. Even though highly specific drugs targeting BRAF mutations have improved overall response rate (ORR) and overall survival (OS) in the last decade, resistances still arise [8]. Generally, BRAF mutations are considered early oncogenic events and are therefore present in most tumor cells, resulting in a low intratumoral heterogeneity [1,44,45]. However, Chiappetta et al. used laser capture microdissection and showed that BRAF mutations cannot be detected in all areas of melanoma, indicating that therapy might not always be equally effective in patients [46]. All four established MUG Mel3 melanoma cell lines, as well as the tumor tissues, exhibited high VAFs for the BRAF V600E allele. Pregnant patients, harbouring a BRAF V600E mutation—the most abundant mutation in melanoma—were described as developing extensive metastatic melanoma during pregnancy or in the first year after birth, with poor survival regardless of therapeutic interventions [47].

Even though new interventions like targeted- and immunotherapies improved OS and progression-free survival (PFS), these therapies involve the risk of severe adverse events and are therefore often contraindicated during pregnancy [42,48].

New treatment strategies with minimal side effects are sought. We tested anti-tumor peptides derived from the human lactoferricin, a host defence peptide targeting negatively charged PS. PS, normally present on the inner leaflet of non-neoplastic cells, was proven to be transferred to the outer leaflet of MUG Mel3 cells, making them suitable for anti-tumor peptide treatment. In previous studies, both tested peptides R-DIM-P-LF11-322 and R-DIM-P-LF334 were shown to exert anti-tumoral activity against melanoma cells [17,18,20,21]. Both tested peptides were active against all four MUG Mel3 cell lines. R-DIM-P-LF11-322, previously described as being more active than R-DIM-P-LF11-334, exhibited lower IC50 values. Regarding serum supplementation, cells grown in the presence of FBS illustrated higher IC50 values than hPL cultured cells, indicating higher amounts of bivalent ions like Ca^2+^ present in FBS media. Bivalent ions were described as competing for negative charges present on the cell surface, lowering the cytotoxic capacity of peptides and thereby delaying the induction of cell death. Furthermore, R-DIM-P-LF11-322 was shown to be less affected by degradation excited by components present in serum [18], and exhibited lower IC50 values. Such new treatment approaches, targeting a common shared feature like PS exposure on the outside of cancerous cells, could help improve PFS by targeting subpopulations responsible for melanoma recurrence after conventional therapies. 

All these findings illustrate the importance of maintaining melanoma heterogeneity to study new treatment options or combination therapies, and to further investigate resistance formation and therapy responses ex vivo.

## 4. Material and Methods

### 4.1. Patient History

A 26-year-old woman suffered from an ulcerated melanoma (tumor thickness 3.9 mm AJCC 2017) in January 2017 on her right thigh. Re-excision had been conducted in February 2017 without sentinel lymph node biopsy in an external hospital. The patient was referred to our university hospital initially in June 2017. The tumor board decided to perform adjuvant treatment with interferon-alpha, and the drug was given from July 2017 till December 2017. Then, the patient stopped due to side effects. In May 2018, she presented with a pathologic lymph node in the right groin and a solitary metastasis was excised in June 2018 (pN1b). At this point, adjuvant treatment was proposed to the patient, but was refused. In November 2018 she presented with pathologic lymph nodes distal to the right groin, while pregnant in the 13th week, which turned out to be PALNM (Pregnancy-associated lymph node metastasis). According to the tumor board, surgery was recommended and scheduled. Melanoma metastases were excised, and the patient was upgraded to pN3b, taking the former metastasis into account. The herein presented study was approved by the local ethics committee board of the Medical University of Graz (vote #31-457ex18/19; valid until 31.07.2020) in accordance with the Helsinki Declaration. The patient gave written informed consent for the study specific procedure. We confirm that all experiments were performed in accordance with relevant guidelines and regulations.

### 4.2. Pooled Human Platelet Lysate (hPL)

Informed consent was obtained from the donors and their blood was routinely tested according to Austrian and European Guidelines. hPL was produced as described [49]. In brief, four buffy coat units of blood group 0 were pooled with one plasma unit of blood group AB (male donor) to prevent interactions of cultured cells with AB0 antigens and isoagglutinins. The pools were centrifuged (461 g, 10 min, 22 °C) to get one unit of platelet-rich plasma (PRP). After leukodepletion by inline filtration, the PRP was frozen for at least 24 h (h) at −30 °C. A total of 10–13 units each were thawed at 37 °C and further pooled to one hPL batch. This summing-up of 50–65 donations achieves a minimization of individual donor variations. The pooled hPL was again frozen and stored at −30 °C. Before final proportioning, pooled hPL was centrifuged at 4000× *g* for 15 min at low brake speed to deplete any immunogenic platelet membrane fragments, and the supernatant was aliquoted and frozen at −30 °C until use. During the manufacturing process, two tests for microbiologic sterility (BacT/ALERT, bioMérieux, France) and mycoplasma contamination were performed.

### 4.3. Cell Culture

A part of the excised lymph nodes was available for cell culture. The material was morphologically divided into a pigmented (P) and non-pigmented (NP) part. Both parts were incubated for 10 min in a 10× concentrated penicillin (100 U/mL)/streptomycin (100 U/mL; pen-strep; Gibco, Life Technologies, Darmstadt, Germany) antibiotic bath in PBS to avoid possible contamination. Afterwards, the pieces were mechanically crushed and divided into different wells and cultivated with FBS or hPL. Four approaches with the following nomenclature were propagated as separate cell lines: MUG Mel3 PF (pigmented lymph node-part cultured in FBS), MUG Mel3 Ph (pigmented lymph node-part cultured in hPL), MUG Mel3 NPF (non-pigmented lymph node-part cultured in FBS), and MUG Mel3 NPh (non-pigmented lymph node-part cultured in hPL). Basic medium was RPMI (Gibco, Life Technologies, Darmstadt, Germany) supplemented with 10% FBS (M&B Stricker, Bernried, Germany) or 10% hPL (obtained from the Department of Blood Group Serology and Transfusion Medicine, Medical University of Graz), 2 mM L-glutamine (Gibco, Life Technologies, Darmstadt, Germany) and for the first passages, additional 1× pen-strep. Complete 10% hPL medium was prepared by addition of 2mM L-glutamine and 10% hPL to RPMI and was left at room temperature (RT) for 4 h. The formed clot was removed, and medium was incubated overnight (O/N) at 4° C. The next day, the medium was aliquoted into 50 mL tubes and warmed up to 37 °C in a water bath for 1 h, then centrifuged for 10 min, 3000× *g* at RT. Medium was filtered through a 0.22 µm filter unit (TPP, Biomedica, Trasadingen, Switzerland). After growth of the cells, an attempt was made to reduce hPL concentrations from 10% to 2.5% (0%; 1.25%; 2.5%; 5%; 7.5%; 10%). hPL cultivated cell lines were frozen in animal-origin free Synth-a-Freeze™ Cryopreservation Medium (Gibco, Life Technologies, Darmstadt, Germany). Cell lines between passages 0–25 were used for experiments.

The *BRAF* V600E mutated melanoma cell line A375 was used as a control and was cultured in DMEM supplemented with 10% FBS, 2 mM L-glutamine, and 1× pen-strep.

All cell lines were periodically checked for mycoplasma contamination (Minerva Biolabs, Berlin, Germany). Cell line authentication was conducted by short tandem repeat (STR) profiling (Promega, Madison, WI, USA; Appendix A). Cell culture microscopic pictures were taken at RT on an Eclipse Ti2 inverted microscope (Nikon, Tokyo, Japan), 10× magnification, numerical aperture 0.30 with a DS-Fi2 camera (Nikon, Tokyo, Japan). Pictures were analysed with the NIS-Elements BR 5.02.00 software (Nikon, Tokyo, Japan). 

### 4.4. Cell Line Authentication

DNA from tumor tissue and cells was prepared using the QIAamp DNA Mini kit (Qiagen, Hilden, Germany) in accordance with the manufacturer’s protocol. After normalizing the DNA, 1 μL of each sample was amplified using the Power Plex^®^ 16 HS System (Promega). One μL of the product was mixed with Hi-Di formamide (Applied Biosystems Inc., Foster City, CA, USA) and Internal Lane Standard (ILS600), denatured and fractionated on an ABI 3730 Genetic Analyzer (Applied Biosystems Inc.). The resulting data were processed and evaluated using ABI Genemapper 3.7.

### 4.5. CellTiter 96^®^ AQueous Non-Radioactive Cell Proliferation Assay (MTS)

To determine optimal hPL concentration in growth medium, MUG Mel3 Ph and MUG Mel3 NPh cells were each seeded at 10^4^ cells/100 µL medium in a clear 96-multiwell plate and incubated for 72 h at 5% CO_2_ and 37 °C. Twenty µL of PMS/MTS solution (Promega) was added to each well. After 2 h of incubation at 5% CO_2_ and 37 °C, OD values were measured at 490 nm and 720 nm for correction wavelength on a CLARIOstar photometer (BMG Labtech, Ortenberg, Germany). This experiment was performed only once in six replicates and was conducted to define optimal hPL concentration for culturing MUG Mel3 cells.

### 4.6. Immunocytochemistry (ICC)

Specific melanoma cell markers were selected for the characterization of isolated primary melanoma cells. Cells were seeded in glass chamber slides (7 × 10^4^ per chamber) and grown in complete medium (10% FBS) to a confluency of 80%. Before fixation, cells were washed with 1× Hanks Balanced Salt Solution (HBSS, Thermo Fisher Scientific, Waltham, MA, USA) and dried for 1 h at RT. Cells were fixed by consecutive wash steps with 4% formaldehyde (Donauchem, Vienna, Austria), 1× PBS (Medicargo, Uppsala, Sweden), ice cold methanol (Merck, Darmstadt, Germany), and ice-cold acetone (Merck). All following procedure steps were performed at RT. All slides were rehydrated in 1× TBE pH 7.5 with 0.1% Tween (Sigma, St. Lois, MO, USA) buffer for 3 min, which was also used as washing buffer in-between steps. Endogenous peroxide activity was quenched by hydrogen peroxide block (Thermo Scientific, Rockford, IL, USA) for 10 min. Ultra V Block from Thermo Scientific was applied to reduce nonspecific background staining, by 5 min incubation. Selected monoclonal primary antibodies were diluted as shown in Appendix A. Negative controls of the same isotype (Dako, Glostrup, Denmark) were applied and incubated for 1 h. After washing, primary enhancer was added for 30 min, followed by HRP Polymer (Thermo Scientific) and AEC chromogen (Thermo Scientific) incubation for 15 min in the dark, and for 5 min, respectively. Finally, cells were counterstained with hematoxylin and blued with running hot tap water each for 5 min. Slides were mounted with aquatex (Merck) and analysed by an Olympus BX53 reflected-light microscope (Hamburg, Germany), 10× magnification, numerical aperture: 0.30 with an Olympus U-TV1X-2 T7 camera (Tokyo, Japan) at RT. Pictures taken were evaluated by the CellSens Standard 2 software (Olympus, Hamburg, Germany). 

### 4.7. qPCR

Melanoma cells (2.2 × 10^5^ cells) were seeded onto six-well plates and incubated for 24 h. Subsequently, RNA isolation was performed with the Monarch^®^ total RNA Miniprep Kit (New England Biolabs, Ipswich., MA, USA) according to the product manual. RNA quality and quantity were measured with a NanoDrop 2000 (Thermo Fisher Scientific). To remove melanin, RNA samples were further filtered using the OneStep PCR Inhibitor Removal Kit (Zymo Research, Irvine, CA, USA). cDNA was obtained from mRNA using the High-Capacity cDNA Reverse Transcription Kit (Thermo Fisher Scientific, Waltham, MA, USA) according to the manufacturer’s manual. qPCR runs were performed on the CFX384 (Bio-Rad Laboratories Inc, Hercules, CA, USA) using primers against Melan-A and Sry-related HMG-Box gene 10 (SOX10; Appendix A), and the 5X HOT FIREPol EvaGreen qPCR Supermix (Solis BioDyne, Tartu, Estonia). Results were analysed using 2^−(ΔΔCt)^ values. Means of GAPDH and β-Actin (ACTB) as reference genes were used for calculation of ΔCt values. Three independent biological experiments were conducted each in triplicates.

### 4.8. FACS Analysis

1 × 10^6^ cells each were harvested from 80–90% confluent cultures and stained in 100 µL FACS buffer with 2.5 µL V450 mouse anti-Human CD271 antibody Clone C40-1457 (BD Horizon, Heidelberg, Germany) for 30 min at 4 °C. Cells were measured on a CytoFLEX S Flow Cytometer (Beckman Coulter, IN, USA) and further analysed using the Software CytExpert 2.3 (Beckman Coulter). The percentage of CD271 positive cells was calculated based on unstained controls and FSC/SSC gating was used to exclude debris, dead cells and doublets. The A375 cell line served as a positive control and Hela cells as negative controls (data not shown). Experiments were run in three independent experiments.

### 4.9. Growth Factors xMAP^®^ Technology

Cytokine concentrations were determined from the non-pigmented lymph node part, using analyte-specific capture beads coated with target-specific capture antibodies according to the manufacturer’s specifications. The analytes were detected by biotinylated analyte-specific antibodies. Following binding of the fluorescent detection label, the reporter fluorescent signal was measured with the Bio-Plex 200 multiplex suspension array system (Bio-Rad, Hercules, CA, USA) and detected with Bio-Plex 5.0 Software (Bio-Rad, Hercules, CA, USA). The sensitivity for the respective cytokines was as followed: CCL1/I-309 1.73–1260 pg/mL; CCL5/RANTES 8.27–6030 pg/mL; CCL11/Eotaxin 20.23–14,750 pg/mL, CXCL1/GRO alpha 15.30–11,160 pg/mL; IL-8/CXCL8 1.44–1050 pg/mL; CCL2/JE/MCP-1 11.87–8650 pg/mL; CCL8/MCP-2 4.66–3400 pg/mL; CCL17/TARC 30.62–22,320 pg/mL; CXCL10/IP 0.43–310 pg/mL. Analyte levels of complete growth medium mixed with the additives FBS or hPL served as background controls (Appendix A). 

### 4.10. DNA Extraction

Genomic DNA of the primary tumor and all four MUG Mel3 cell lines was isolated on a Maxwell MDxResearch System (Promega).

### 4.11. Copy Number Profiling

Genome-wide copy number alterations (CNA) were established using shallow whole-genome sequencing (sWGS). Shotgun libraries were prepared using the TruSeq DNA LT Sample preparation Kit (Illumina, San Diego, CA, USA). Briefly, 380 ng, 144 ng and 360 ng input DNA from all four MUG Mel3 cell lines and both parts of the lymph node were fragmented in 130 µL using the Covaris System (Covaris, Woburn, MA, USA). After concentrating the volume to 50 µL, end repair, A-tailing and adapter ligation were performed following the manufacturer’s instructions. For selective amplification of the library fragments that have adapter molecules on both ends, 15 PCR cycles were used for higher concentration samples. Libraries were quality checked on an Agilent Bioanalyzer using a DNA 7500 Chip (Agilent Technologies, Santa Clara, CA, USA) and quantified using qPCR with a commercially available PhiX library (Illumina, San Diego, CA, USA) as a standard. Libraries were pooled equimolarily and sequenced on an Illumina MiSeq in a 150 bp single read run. On completion of the run data were base-call demultiplexed on the instrument (provided as Illumina). Genome-wide copy number calling was performed as previously described [50].

### 4.12. Mutation Analysis Using AmpliSeq (BRAF)

Highly multiplexed PCR was used to generate amplicon libraries to amplify 207 amplicons, covering approximately 2800 COSMIC mutations from 50 oncogenes and tumor suppressor genes. (Cancer Hotspot Panel v2, Cat. Nr. 4475346; Thermo Fisher Scientific, Waltham, MA, USA). All analyses were performed in duplicate. Libraries were prepared using the Ion AmpliSeq Library Kit 2.0 and sequencing was performed on an Ion Proton Sequencer (Thermo Fisher Scientific, Waltham, MA, USA). Emulsion PCR and sequencing runs were performed with the appropriate kits (Ion One Touch Template Kit version 2 and Ion Proton 200 Sequencing Kit; Thermo Fisher Scientific) using Ion PI chips. Sequencing length was set to 520 flows and yielded reads ranging from 70 to 150 bp, consistent with the expected amplicon size range. Initial data analysis was performed using the Ion Torrent Suite Software version 4.1 Plug-ins (Thermo Fisher Scientific).

### 4.13. Tumorigenicity Study

CR ATH HO mice (Crl:NU(NCr)-Foxn1nu, Charles River Laboratories, Kent, UK) and NXG mice (NOD-Prkdcscid-IL2rgTm1/Rj, Janvier Labs, Saint Berthevin Cedex, France) were maintained in-house (4–5 weeks of age, weight between 15 and 20 g). In accordance with a protocol approved by the committee for institutional animal care and use at the Austrian Federal Ministry of Science and Research (BMWFW); vote 66.010/0046-WF/V/3b/2016) animal work was carefully carried out. All mice were maintained under specific pathogen free (SPF) conditions in individually ventilated cages with ad libitum access to food and water. For cell injection and ultrasound imaging, mice were anaesthetized with constant administration of 2% isoflurane in a constant airflow of 2.5 L per minute. Animals were sacrificed 25 days post-injection. 

Mice were divided into two groups (*n* = 5) according to strain. Animals were injected into each flank subcutaneously with 2.5 × 10^6^ cells in 100 µL PBS cell suspension using a 27G needle. Injection was performed as described in the following: front right flank received MUG Mel3 NPh, back right flank received MUG Mel3 NPF, front left flank received MUG Mel3 Ph, back left flank received MUG Mel3 PF. Inoculation of the tumor was monitored daily and ultrasound imaging started on day six post-injection once a week. After sacrifice, histopathological examination was performed. The mice were dissected, and tumors extracted. All other organs were checked visually for structural changes. The tissue was fixed in 4% paraformaldehyde solution for 24 h, and were further embedded in paraffin. 

### 4.14. Ultrasound Imaging

High-frequency ultrasound (HF-US) was performed using a Vevo3100 HF-US system with a 50 MHz (MX700) or a 40 MHz (MX550D) transducer (Fujifilm VisualSonics, Inc., Toronto, ON, Canada) reaching a spatial resolution of about 30–40 µm. Images in sagittal and transverse planes were obtained of the region of interest. Calculation of tumor volume was performed using VevoLab Software (Version 5.5.1, Fujifilm VisualSonics, Inc.) using the formula displayed in Equation (1): (1)$$V\;\left[{\mathrm{mm}}^{3}\right]=\;\frac{3.141\times \mathrm{length}\;\left[\mathrm{mm}\right]\times \mathrm{width}\;\left[\mathrm{mm}\right]\times \mathrm{depth}\;\left[\mathrm{mm}\right]}{6}$$

### 4.15. PS Exposure

PS exposure was measured using the RealTime-Glo™ Annexin V Apoptosis Assay (Promega). Briefly, cells were seeded at 10^5^ cells/100 µL medium in a white 96-multiwell plate with clear bottom and incubated O/N at 5% CO_2_ and 37 °C. The reagent stock solution (1000×) was diluted in respective medium according to the manufacturer’s protocol. Before the measurement 100 µL of the reagent solution was added to the cell suspension. Luminescence was measured using the Glomax Multi+ detection system (Promega, Madison, WI, USA).

### 4.16. Peptides

C-terminally amidated peptides R-DIM-P-LF11-322 (PFWRIRIRRPRRIRIRWFP-NH_2_, M = 2677.4 g/mol) and R-DIM-P-LF11-334 (PWRIRIRRPRRIRIRWP-NH2, M = 2382.4 g/mol) derived from Lactoferricin were purchased from PolyPeptide Group (San Diego, CA, USA). A purity of higher than 96% for all peptides had been determined by RP-HPLC. Peptide stock solutions were prepared in acetic acid (0.1%, *v*/*v*) to an approximate concentration of 3 mg/mL and treated by ultrasonication at 15 min for better solubility. Peptide concentration was determined by measurement of UV absorbance of tryptophan at a wavelength of 280 nm using a NanoDrop photometer (ND 1000, Peqlab, VWR International, Inc., Erlangen, Germany). All peptide stocks were stored at 4 °C until use.

### 4.17. PI Uptake Toxicity Assay

To determine peptide-induced cell death, cells were collected, resuspended in respective media and diluted to a concentration of 1 × 10^6^ cells/mL. 100 µL aliquots (10^5^ cells) were incubated with different peptide concentrations (5 µM to 100 µM) for up to 8 h in the presence of propidium iodide (PI; 2 µL/10^5^ cells of 50 µg/mL, Molecular Probes Inc., Eugene, OR, USA) at RT in black 96-well plates. PI uptake was measured using the GloMax^®^ Multi+ Detection System (Promega, Madison, WI, USA). Cytotoxicity was calculated from the percentage of PI-positive cells in media alone (P_0_) and in the presence of peptide (P_X_; Equation (2)). Triton-X-100 was used to determine 100% of PI positive cells (P_100_).
(2)$$\%\;\mathrm{PI}-\mathrm{uptake}=\;\frac{100\times \left(P_{x}-P_{0}\right)}{\left(P_{100}-P_{0}\right)}$$

Excitation and emission wavelengths were 536 nm and 617 nm, respectively. Each experiment was repeated at least 3 times.

### 4.18. Statistical Analysis

Statistical analyses were conducted in GraphPad Prism 9.2.0 (GraphPad Software, San Diego, CA, USA) using two-tailed student’s *t*-tests to compare culture conditions (FBS or hPL) within cell line origins (pigmented or non-pigmented) or to compare the origins of cells within the same cultivation method, and two-tailed unpaired student’s *t*-test to compare tumor weight and tumor growth of the four culture conditions on the day of sacrifice. A *p*-value ≤ 0.05 was considered statistically significant. 

## 5. Conclusions

Melanoma in pregnancy is a vast, debated and complex field. Further research is urgently needed to provide evidence-based treatment and improve the management of melanoma in pregnancy. We have successfully established four cell lines from a melanoma metastasis from a pregnant woman. By using different tumor fractions and cultivation conditions, we were able to better mimic intratumoral heterogeneity as the four cell lines differ morphologically, genetically and phenotypically. In addition, we were able to demonstrate different cytotoxic effects by antitumor peptide treatments for all four cell lines, and thus may also consider them for new innovative treatment strategies.

## Figures and Tables

**Figure 1 ijms-22-11318-f001:**
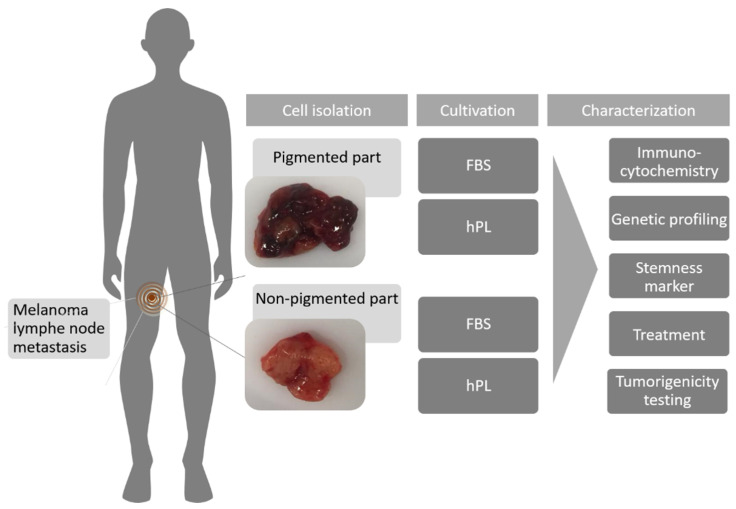
Overview of the experimental procedures. The location of the lymph node metastasis is illustrated, as well as the morphology of both areas (pigmented and non-pigmented). Furthermore, information about the cultivation conditions and individual steps in characterization are provided.

**Figure 2 ijms-22-11318-f002:**
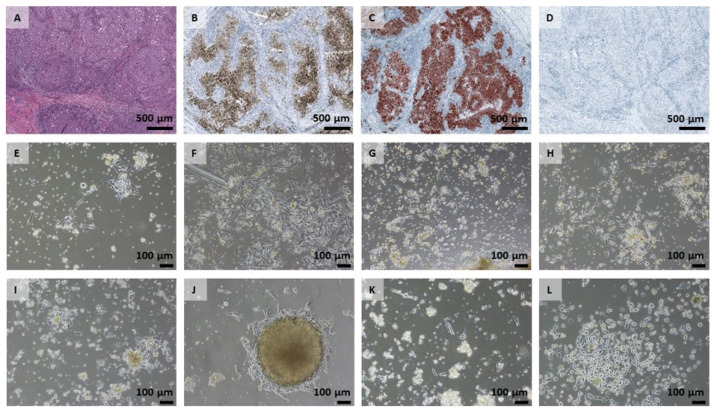
IHC of primary tumor tissue and morphological overview of cell growth. (**A**–**D**) Formalin fixed, paraffin embedded (FFPE) and stained tissue sections of the primary tumor tissue. (**A**) H&E, (**B**) HMB45; (**C**) Melan-A; (**D**) Tyrosinase. Stained tissue sections were scanned with a PANNORAMIC^®^ 1000 (3DHISTECH, Budapest, Hungary) and pictures were analysed using CaseViewer 2.4 software (3DHISTECH, Budapest, Hungary). (**E**–**L**) Morphological presentation of the individual cultures observed over time; representative areas are illustrated. (**E**) MUG Mel3 PF (day six); (**F**) MUG Mel3 PF (day eight); (**G**) MUG Mel3 Ph (day six); (**H**) MUG Mel3 Ph (day eight); (**I**) MUG Mel3 NPF (day six); (**J**) MUG Mel3 NPF (day eight); (**K**) MUG Mel3 NPh (day six); (**L**) MUG Mel3 NPh (day eight).

**Figure 3 ijms-22-11318-f003:**
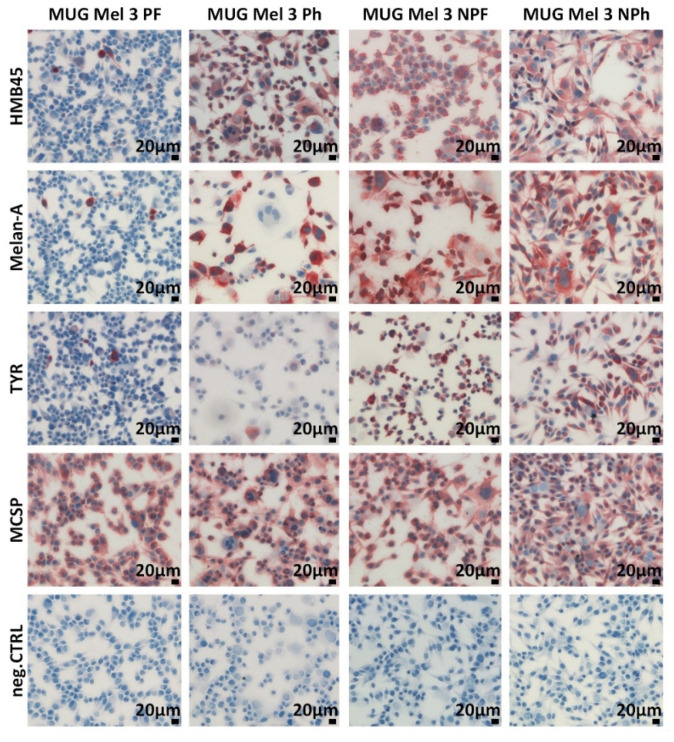
ICC for melanoma cell markers of cultivated MUG Mel3 cell lines. HMB45 staining: MUG Mel3 PF weak staining, MUG Mel3 Ph, MUG Mel3 NPF, and MUG Mel3 NPh strong staining. Melan A: MUG Mel3 PF weak staining, MUG Mel3 Ph, MUG Mel3 NPF, and MUG Mel3 NPh strong staining. Tyrosinase: MUG Mel3 PF and MUG Mel3 Ph weak staining, MUG Mel3 NPF and MUG Mel3 NPh strong staining. MCSP: highly expressed in all four cell lines. For each tested antigen, the corresponding IgG isotype control (negative control) was applied.

**Figure 4 ijms-22-11318-f004:**
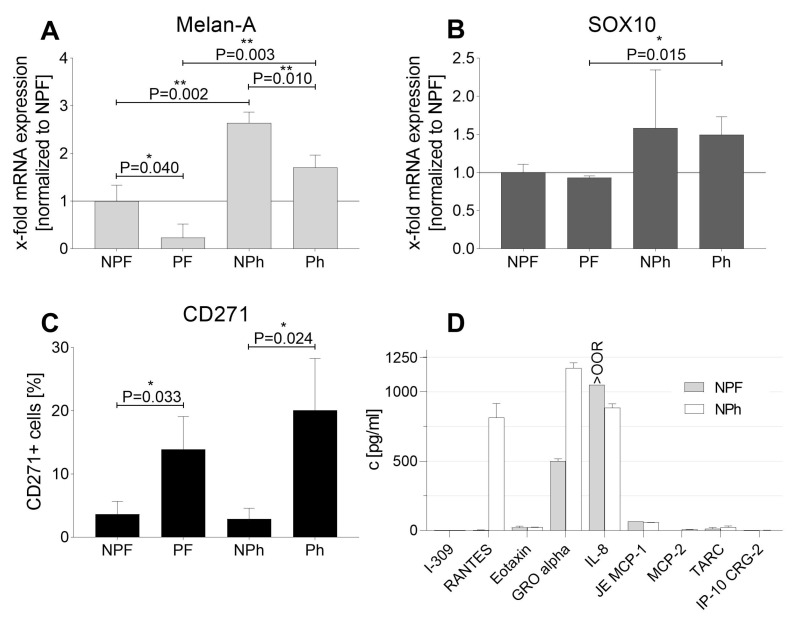
Melan-A and SOX10 mRNA, and percentage of CD271 expression levels as well as chemokine secretion in MUG Mel3 cultures. (**A**) Melan-A mRNA expression in all used cell lines, comparison normalized to MUG Mel3 NPF culture (*n* = 3); (**B**) SOX10 mRNA expression in all four cell lines, comparison normalized to MUG Mel3 NPF (*n* = 3) (**C**) CD271 surface marker expression determined by FACS analyses (*n* = 3); (**D**) secretion of chemokines in different culture conditions using FBS and hPL. The statistical analyses were conducted in GraphPad Prism 9.2.0 using a two-tailed student’s t-tests to compare culture methods (FBS and hPL) within cell line origins (pigmented or non-pigmented) or to compare the origins of cells within the same cultivation method. A *p*-value ≤ 0.05 was considered statistically significant (*p* ≤ 0.05 = *, *p* ≤ 0.01 = **).

**Figure 5 ijms-22-11318-f005:**
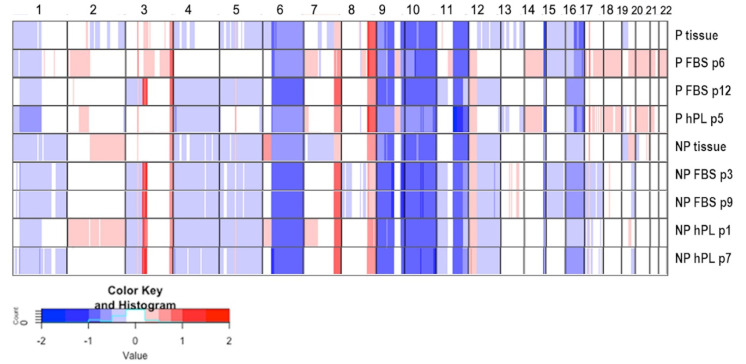
Genome-wide copy number profile of lymph node tissues and corresponding cell lines. Heat map depicting segmented log2-ratios pigmented (P) and non-pigmented (NP) tissues and various passages of their corresponding cell lines cultivated with FBS or hPL. Blue indicates loss of chromosomal material; red indicates gain of chromosomal material.

**Figure 6 ijms-22-11318-f006:**
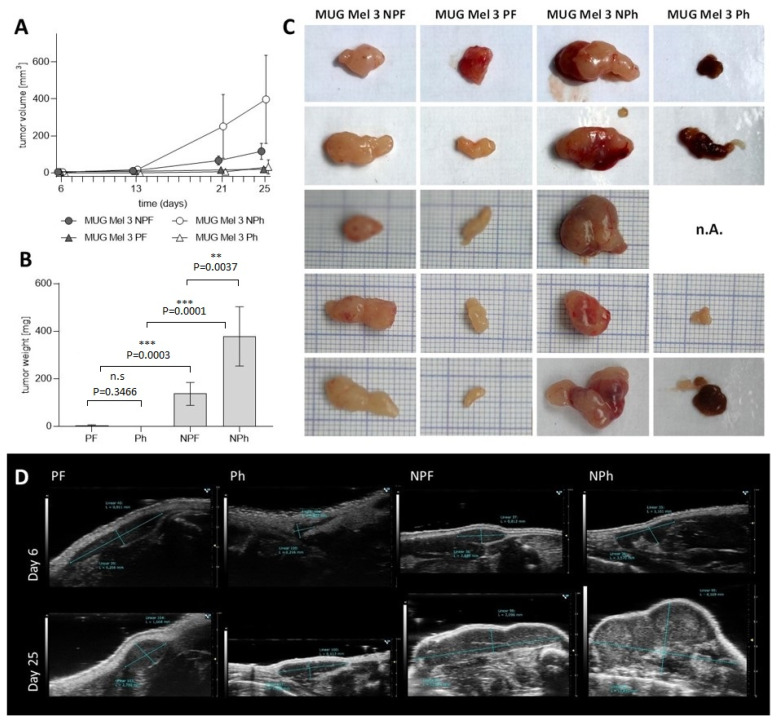
Tumorigenic profile of all four MUG Mel3 cell lines in CR ATH HO nude mice. Tumor volume [mm^3^] over time—significance was tested by comparing culture conditions (FBS or hPL) within cell lines of origin (pigmented or non-pigmented) and the origins of cells within the same cultivation method on day of sacrifice: MUG Mel3 NPh/MUG Mel3 NPF *p*-value = 0.0319; MUG Mel3 NPh/ MUG Mel3 Ph *p*-value = 0.0056; MUG Mel3 PF/Ph *p*-value = 0.5024 and MUG Mel3 NPF/ MUG Mel3 PF *p* value = 0.0011—(**A**) and tumor weight [mg] on day of sacrifice (**B**) are illustrated (*p* ≤ 0.01 = **, *p* ≤ 0.001 = ***). Excised tumors of all mice are shown in (**C**) (*n* = 5). High frequency ultrasound (HF-US) images of one representative mouse are presented in (**D**). The images on day six were recorded with a 50 MHz (MX700) transducer; the images on day 25 with a 40 MHz (MX550D) transducer.

**Figure 7 ijms-22-11318-f007:**
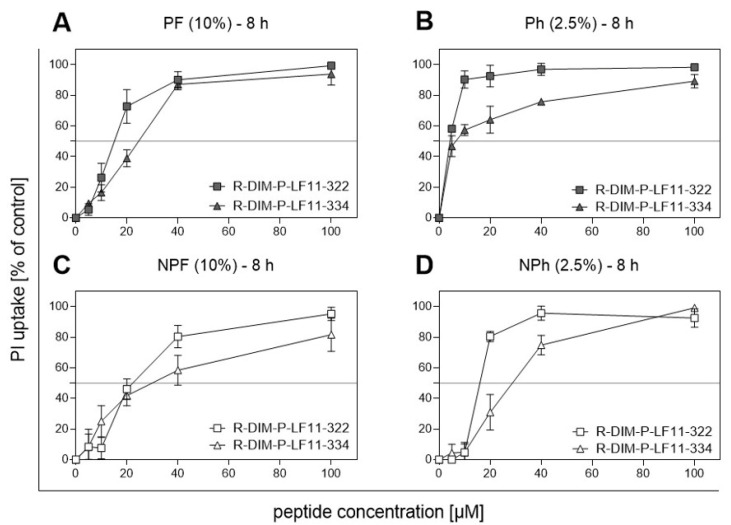
Cytotoxic capacity of R-DIM-P-LF11-322 and R-DIM-P-LF11-334 peptides on all four MUG Mel3 cell lines. Cytotoxicity was measured by PI-uptake in the presence of 10% FBS or 2.5% hPL after 8 h of incubation. (**A**) MUG Mel3 PF, (**B**) MUG Mel3 Ph, (**C**) MUG Mel3 NPF, and (**D**) MUG Mel3 NPh. Data for PI measurements from at least three independent experiments are presented as mean ± SD.

**Table 1 ijms-22-11318-t001:** Overview of melanoma marker. ICC markers HMB45, Melan-A, tyrosinase, and MCSP, and mRNA levels for SOX10, Melan-A, and % CD271 are summarized for all four cell lines. ICC data are depicted in Figure 3, qPCR data in Figure 4. For normalization of qPCR data, GAPDH, and β-Actin were used as housekeeping genes. qPCR and CD271 levels were obtained from three independent experiments.

	Pigmented		Non-Pigmented	
	FBS	hPL	FBS	hPL
Growth Behavior	Fast	Slow	Fast	Slow
Adhesion	++	Semi-adherent	++	Semi-adherent
ICC-HMB45	−	+	++	++
ICC-Melan-A	−	+	++	++
ICC-tyrosinase	−	−	+	++
ICC-MCSP	++	++	++	++
qPCR-Sox10	+	++	+	++
qPCR-Melan-A	−	++	+	++
CD271	13.94% ± 4.22	20.09% ± 6.71	3.66% ± 1.67	2.91% ± 1.40

**Table 2 ijms-22-11318-t002:** Overview of melanoma markers expressed in xenografts.

CR ATH HO	MUG Mel3 PF	MUG Mel3 Ph	MUG Mel3 NPF	MUG Mel3 NPh
HMB45	++	++	++	++
Melan-A	+	++	++	++
Tyrosinase	−	+	−	+
**NXG**	**MUG Mel3 PF**	**MUG Mel3 Ph**	**MUG Mel3 NPF**	**MUG Mel3 NPh**
HMB45	+	++	++	++
Melan-A	+	++	++	++
Tyrosinase	−	+	−	−

**Table 3 ijms-22-11318-t003:** IC_50_ values [µM] of peptides R-DIM-P-LF11-322 and R-DIM-P-LF11-334 on all four MUG Mel3 cell lines. IC_50_ ± SD [µM] is summarized for all MUG Mel3 cell lines cultivated in 2.5% hPL or 10% FBS. *p*-values were calculated between the effects of R-DIM-P-LF11-322 and R-DIM-P-LF11-334.

	IC_50_ (PI) [µM]		
	R-DIM-P-LF11-322	R-DIM-P-LF11-334	*p*-Value
MUG Mel 3 PF (10% FBS)	14.4 ± 0.4	21.9 ± 1.7	0.0017
MUG Mel 3 Ph (2.5% hPL)	4.3 ± 0.3	6.6 ± 0.7	0.0064
MUG Mel 3 NPF (10% FBS)	22.1 ± 1.2	28.1 ± 1.3	0.0042
MUG Mel 3 NPh (2.5% hPL)	16.0 ± 0.9	26.9 ± 0.7	0.0001

## Data Availability

Not applicable.

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
