# Peer review of "MUG Mel3 Cell Lines Reflect Heterogeneity in Melanoma and Represent a Robust Model for Melanoma in Pregnancy"

_ijms, 2021, doi:10.3390/ijms222111318_

Round 1

Reviewer 1 Report

The manuscript by Schrom et al. describes the generation and characterization of cell lines to be used as a model of pregnancy-associated melanoma. Due to its occurrence at younger ages, melanoma is one of the malignancies that most frequently take place concomitantly with pregnancy. Therefore, in vitro models are strongly required. The authors described the generation of 4 different cell lines based on the part of the tissue from which the sample was obtained and on the culture with FBS or platelet lysate. The cell lines are characterized using different methods. In general, the study design is appropriate. Although the trends shown in most of the graphs would support the conclusions raised by the authors, statistical tests are missing for some of the results (see below). The following aspects should be addressed before publication:

  1. Supplementary table 1. STR profiling. The upper part of the table lacks a heading specifying the loci investigated.
  2. Table 1 shows no differences for Sox10 expression between the 4 groups. However, fig 4B depicts some significant differences between groups. Please, check.
  3. The statistical significance of results presented in lines 227-230 and depicted in figure 6 A and B has to be analysed. Results from supplementary figures 3 A and B and 5 also need to be analysed statistically.
  4. Statistical analysis of cytotoxicity data must be performed in order to confirm the claims on peptide efficacy and cytotoxicity.
  5. Line 366. Please, check the spelling of “May”.
  6. Section 4.7. Line 480. Please, specify what you consider as 3 independent experiments (3 different cell cultures? 3 different RNA isolations?)
  7. Sections 4 and 5 should be merged, as both correspond to materials and methods.
  8. Figure 1. What do you mean with “CNV”? Since the abbreviation has not been defined before in the manuscript, it should be defined in the figure legend.

Reviewer 2 Report

The authors isolated 4 cell lines from

pregnant patient with melanoma. The

authors performed molecular biology and

genomics assays to characterize melanoma cells.

There are several points

  1. please include asterisk with p-value
  2. to Figures. 
  3. use word cytokines instead of c on
  4. Y-axis with cytokine assays Figure
  5. 2.6 part, check first sentence add in vivo rather than in

Round 2

Reviewer 1 Report

The authors provided a significantly improved version where most of my concerns where addressed. 

The only point which still requires attention is Figure 7, supplementary figure 5 and results from section 2.6. Several claims presented in this section cannot be made without a proper statistical analysis.

For instance, the authors state:  "Cells cultivated with FBS presented significant more PS on the outside of cells in both cultured parts, pigmented and non-pigmented (Supplementary figure 5)". However, no test to assess statistical significance was performed.

The same goes for the peptide related cytotoxicity. In lines 307-312, clear comparisons between peptides and cells are made.  The authors should either perform a statistical analysis or clearly state in the manuscript why a statistical analysis does not need to be performed.

Author Response

Comments and Suggestions for Authors

The authors provided a significantly improved version where most of my concerns where addressed. 

The only point which still requires attention is Figure 7, supplementary figure 5 and results from section 2.6. Several claims presented in this section cannot be made without a proper statistical analysis.

For instance, the authors state:  "Cells cultivated with FBS presented significant more PS on the outside of cells in both cultured parts, pigmented and non-pigmented (Supplementary figure 5)". However, no test to assess statistical significance was performed.

Comment to the reviewer: Sorry for the missing statistic. We have now improved the suppl figure 5 and adjusted the figure legend.

Supplementary figure 5: Expression of PS exposure measured by luminescence. PS exposure was measured using the RealTime-Glo™ Annexin V Apoptosis Assay from Promega. Measured PS levels of MUG Mel 3 were statistically significant lower in hPL containing medium compared to FBS. Experiments were performed at least three times. RLU: relative Luminescence.

Furthermore we added in section 2.6: Cells cultivated with FBS presented significant (PF/ NPF: p-value=0.0250; PF/Ph p-value=0.0002; NPh/Ph: p-value=0.0887 and finally NPF/ NPh p-value=0.0013) more PS on the outside of cells in both cultured parts, pigmented and non-pigmented (Supplementary figure 5).

The same goes for the peptide related cytotoxicity. In lines 307-312, clear comparisons between peptides and cells are made.  The authors should either perform a statistical analysis or clearly state in the manuscript why a statistical analysis does not need to be performed.

Comment to the reviewer: A statistical analysis (student’s t test) between peptides and origin of cells has been integrated into the results (see table 3), where applicable.

Table 3: IC50 values [µM] of peptides R-DIM-P-LF11-322 and R-DIM-P-LF11-334 on all four MUG Mel3 cell lines. IC50 ± SD [µM] is summarized for all MUG Mel3 cell lines cultivated in 2.5% hPL or 10% FBS. P-values were calculated between the effect of R-DIM-P-LF11-322 and R-DIM-P-LF11-334.

IC50 (PI) [µM]

R-DIM-P-LF11-322

R-DIM-P-LF11-334

P-value

MUG Mel 3 PF (10% FBS)

14.4 ± 0.4

21.9 ± 1.7

0.0017

MUG Mel 3 Ph (2.5% hPL)

4.3 ± 0.3

6.6 ± 0.7

0.0064

MUG Mel 3 NPF (10% FBS)

22.1 ± 1.2

28.1 ± 1.3

0.0042

MUG Mel 3 NPh (2.5% hPL)

16.0 ± 0.9

26.9 ± 0.7

0.0001

Changes in the text have been made accordingly (page 19):  Comparing both peptides, R-DIM-P-LF11-322 elicited cytotoxic effects at lower concentrations than R-DIM-P-LF11-334 ranging from 4.3 ± 0.3 µM for MUG Mel3 Ph up to 22.1 ± 1.2 µM for MUG Mel3 NPF. When focusing on the origin of cells, IC50-values for R-DIM-P-LF11-322 and R-DIM-P-LF11-334 on cells cultured from the non-pigmented part of the lymph node metastasis were significant higher compared to cells from the pigmented part, as analyzed by the student´s t test (p<0.001) expect the effect from R-DIM-P-LF1134 in PF/NPF (p<0.01). In detail the effect of R-DIM-P-LF11-322 compared to all cell lines: PF/Ph p-value=0.0001,  PF/NPF p-value=0.0005; NPF/NPh p-value=0.0021 and NPh/Ph p-value=0.0001 and in detail the effect of R-DIM-P-LF11-334 compared to all cell lines: PF/Ph p-value= 0.0001; PF/NPF p-value=0.0074, NPF/NPh p-value=0.232, NPH/Ph p-value=0.0001 (Figure 7 A – D).

Round 3

Reviewer 1 Report

The authors have addressed all my concerns.